# Interplay between IL-10, IFN-γ, IL-17A and PD-1 Expressing EBNA1-Specific CD4^+^ and CD8^+^ T Cell Responses in the Etiologic Pathway to Endemic Burkitt Lymphoma

**DOI:** 10.3390/cancers13215375

**Published:** 2021-10-27

**Authors:** Catherine S. Forconi, David H. Mulama, Priya Saikumar Lakshmi, Joslyn Foley, Juliana A. Otieno, Jonathan D. Kurtis, Leslie J. Berg, John M. Ong’echa, Christian Münz, Ann M. Moormann

**Affiliations:** 1Division of Infectious Diseases and Immunology, Department of Medicine, University of Massachusetts Chan Medical School, Worcester, MA 01605, USA; Catherine.Forconi@umassmed.edu (C.S.F.); Priya.SaikumarLakshmi@umassmed.edu (P.S.L.); Joslyn.foley@cshs.org (J.F.); 2Center for Global Health Research, Kenya Medical Research Institute, Kisumu 40100, Kenya; dmulama@mmust.ac.ke (D.H.M.); michaelongecha@gmail.com (J.M.O.); 3Jaramogi Oginga Odinga Teaching and Referral Hospital, Ministry of Medical Services, Kisumu 40100, Kenya; junyarchiga@gmail.com; 4Center for International Health Research, Department of Pathology and Laboratory Medicine, Rhode Island Hospital, The Warren Alpert Medical School of Brown University, Providence, RI 02903, USA; Jonathan_Kurtis@brown.edu; 5Department of Immunology and Microbiology, University of Colorado, Anschutz Medical Campus, Aurora, CO 80045, USA; LESLIE.BERG@UCDENVER.EDU; 6Department of Viral Immunobiology, Institute of Experimental Immunology, University of Zürich, CH-8057 Zurich, Switzerland; christian.muenz@uzh.ch

**Keywords:** endemic Burkitt lymphoma, EBV, T-cells, EBNA1, cytokines, PD-1, malaria, Kenya

## Abstract

**Simple Summary:**

Endemic Burkitt lymphoma (eBL) is a common pediatric cancer in sub-Saharan Africa. The incidence of this aggressive B-cell cancer is linked to *Plasmodium falciparum (Pf)* malaria and Epstein–Barr virus (EBV) co-infections during childhood. Most eBL tumors contain EBV and are characterized by the Epstein–Barr Nuclear Antigen 1 (EBNA1) latency I pattern of viral gene expression. The aim of our study was to compare the phenotypes and functions of CD4^+^ and CD8^+^ T cell responses to EBNA1 in children diagnosed with eBL and in healthy EBV-seropositive children to highlight differences that contribute to the balance between anti-viral immunity and eBL pathogenesis.

**Abstract:**

Children diagnosed with endemic Burkitt lymphoma (eBL) are deficient in interferon-γ (IFN-γ) responses to Epstein–Barr Nuclear Antigen1 (EBNA1), the viral protein that defines the latency I pattern in this B cell tumor. However, the contributions of immune-regulatory cytokines and phenotypes of the EBNA1-specific T cells have not been characterized for eBL. Using a bespoke flow cytometry assay we measured intracellular IFN-γ, IL-10, IL-17A expression and phenotyped CD4^+^ and CD8^+^ T cell effector memory subsets specific to EBNA1 for eBL patients compared to two groups of healthy children with divergent malaria exposures. In response to EBNA1 and a malaria antigen (*Pf*SEA-1A), the three study groups exhibited strikingly different cytokine expression and T cell memory profiles. EBNA1-specific IFN-γ-producing CD4^+^ T cell response rates were lowest in eBL (40%) compared to children with high malaria (84%) and low malaria (66%) exposures (*p* < 0.0001 and *p* = 0.0004, respectively). However, eBL patients did not differ in CD8^+^ T cell response rates or the magnitude of IFN-γ expression. In contrast, eBL children were more likely to have EBNA1-specific CD4^+^ T cells expressing IL-10, and less likely to have polyfunctional IFN-γ^+^IL-10^+^ CD4^+^ T cells (*p* = 0.02). They were also more likely to have IFN-γ^+^IL-17A^+^, IFN-γ^+^ and IL-17A^+^ CD8^+^ T cell subsets compared to healthy children. Cytokine-producing T cell subsets were predominantly CD45RA^+^CCR7^+^ T_NAIVE-LIKE_ cells, yet PD-1, a marker of persistent activation/exhaustion, was more highly expressed by the central memory (T_CM_) and effector memory (T_EM_) T cell subsets. In summary, our study suggests that IL-10 mediated immune regulation and depletion of IFN-γ^+^ EBNA1-specific CD4^+^ T cells are complementary mechanisms that contribute to impaired T cell cytotoxicity in eBL pathogenesis.

## 1. Introduction

Endemic Burkitt lymphoma (eBL), the most common pediatric cancer in equatorial Africa [1,2] has long been associated with early-age infections by two pathogens: Epstein–Barr virus (EBV) and *Plasmodium falciparum* [3,4]. Children at risk of eBL experience primary EBV infection before 2 years of age [5,6,7] and have malaria-associated episodes of viral reactivation resulting in cumulatively higher EBV loads [6,7,8] and IFN-γ deficiencies against EBV lytic and latent antigens [9,10]. Recent studies have shown that the EBV status of eBL tumors is associated with distinct mutational landscapes [11,12,13], supporting the premise that EBV plays a role that transcends tumorigenesis. During early childhood, repeated *P. falciparum* (*Pf*) infections are also common for those residing in holoendemic areas and severe disease is the leading cause of morbidity and mortality in children under 5 years of age [14]. However, eBL is most often diagnosed between 5 and 9 years of age [2], after the development of premunition which is characterized by the ability to harbor asymptomatic and, therefore, untreated persistent *Pf*-malaria infections. Thus, prolonged malaria exposure has been postulated to diminish EBV-specific T cell immune surveillance as a risk factor in eBL pathogenesis [15,16,17,18]. Despite the long-standing epidemiological association, our understanding of how malaria disrupts the balance between antiviral immunity, which is insufficient to eliminate EBV yet sufficient to limit eBL oncogenesis, is incomplete. Furthering our understanding of the underlying mechanisms involved in the etiology of eBL will help identify alterations in immune cells that are important to control EBV. 

Our knowledge of EBV-specific T cell immunity that minimizes its pathologic impact has been gained primarily from studying young adults residing in non-malarious areas, where robust immunity to EBV lytic and latent antigens decreases viral loads to undetectable levels [19,20]. Once this persistent herpesvirus infection is established, EBV evades immune detection by down-regulating viral protein expression during latency [21]. Of particular interest as an immunotherapeutic target is the Epstein–Barr nuclear antigen 1 (EBNA1), which is the viral antigen consistently expressed by all EBV-associated B cell tumors [22], as well as during the lytic and latent cycles of all EBV infections [23]. EBNA1 is recognized by CD4^+^ T cells via macroautophagy [24] but decreases translation and proteasomal processing by major histocompatibility (MHC) class I molecules due to a glycine-alanine repeat domain [25,26] thereby limiting the development of CD8^+^ T cell responses, which are engaged by non-classical MHC class I cross-presentation [27,28,29]. Our studies of children diagnosed with eBL were the first to describe a selective deficiency in IFN-γ responses to EBNA1 [18]. However, this study was limited to measuring IFN-γ by ELISPOT assay which did not determine the phenotype of EBNA1-specific cells or alternative cytokine expression profiles.

Based on these observations, we set forth to explore the influence of prolonged exposure to *Pf* during childhood on the quality of EBNA1-specific T cell immunity. First, we sought to determine if *Pf* co-infections led to the polarization of EBNA1-specific T cell immunity away from Th1 (IFN-γ-producing) towards anti-inflammatory, immune-regulatory T cells (i.e., IL-10 expressing) [30], or the generation of an alternative proinflammatory Th17 (IL-17A) phenotype [31]. For those with EBNA1-specific T cell responses, we compared the phenotype of the cytokine-producing cells to determine if young age or exposure to malaria were associated with distinct T cell subsets. We further explored the potential heterogeneity in human T cell responses, which has been described for many infectious diseases, including malaria [32], and characterized based on the expression of CD45RA and CCR7 to define central memory, effector memory, terminally differentiated RA re-expressing effector memory and naïve-like cell subsets [33]. Finally, other studies reported the upregulated expression of programmed cell-death 1 (PD-1) in virus-specific T cells [34], during malaria infections [35] and other parasitic infections [36,37] as an indicator of T cell exhaustion or persistent antigen stimulation. We speculated that EBNA1-specific T cell exhaustion could be present in children after a prolonged exposure to *Pf* and concomitantly higher EBV loads as a precursor to eBL tumorigenesis.

We conducted a case–control study of children diagnosed with eBL compared to two groups of healthy children residing in areas of Kenya with divergent malaria endemicity. All three cohorts experienced their primary EBV infection before 2 years of age [7]. We determined the frequency and phenotype of CD4^+^ and CD8^+^ T cell subsets across our three study populations, defined by multiparameter flow cytometry (CCR7, CD45RA, PD-1) and intracellular cytokine (IFN-γ, IL-10 and IL-17A) expression performed on peripheral blood mononuclear cells (PBMCs) after the *ex vivo* stimulation with EBNA1. We also compared EBNA1 responses to T cell responses against a malaria-specific *Pf* schizont egress antigen-1A (*Pf*SEA-1A) [38]. Through this study, we show different pathogen-dependent (i.e., EBNA1 or *Pf*SEA-1A) cytokine expressions and T cell memory profiles between our three groups of children.

## 2. Materials and Methods

### 2.1. Study Populations

Children diagnosed with eBL were enrolled at Jaramogi Oginga Odinga Teaching and Referral Hospital, a cancer referral center in Kisumu, Kenya. Diagnosis was confirmed by two independent pathologists and molecular methods [11,39] and blood samples were collected prior to chemotherapy. Healthy children were enrolled from two counties in western Kenya that had divergent *Pf* transmission patterns and, accordingly, differed in their incidence of eBL [40]. Kisumu County, a malaria holoendemic region (i.e., a region with recurrent or persistent malaria infections during childhood as previously described, is referred to as “chronic” malaria exposure) and Nandi County, a hypo-endemic region for malaria (i.e., few “acute” malaria infections during childhood) [41]. Adults from Nandi (*n* = 3) and Kisumu (*n* = 4) were enrolled as controls. Study participants did not have blood stage malaria infections when venous blood samples were collected (diagnosed by blood smear). To confirm differences in EBV load between study groups, viral copies were quantified as previously described [7]. PBMCs were processed within 3 h by Ficoll-Hypaque density centrifugation followed by cryopreservation.

Written informed consent was obtained from adult study participants and the parents or guardians prior to their children’s enrollment in this study. Ethical approval was obtained from the Institutional Review Board at the University of Massachusetts Chan Medical School, Worcester, MA, USA and the Scientific and Ethics Review Unit at the Kenya Medical Research Institute, Nairobi, Kenya.

### 2.2. Characteristics of Study Populations

Study participant characteristics are summarized in Appendix A. Children in the three groups had similar male-to-female ratios and ages (ranging from 2 to 14 years, with a median age of 8). As previously published [6], EBV copies were significantly higher for eBL children compared to detectable viremia in healthy Kisumu (high malaria) and Nandi (low malaria) children (*p* < 0.0001 and *p* < 0.0001, respectively). Resting CD4/CD8 ratios did not differ between study groups; however, Nandi children had fewer CD19^−^CD14^−^CD3^+^ lymphocytes compared to Kisumu and eBL children (*p* = 0.002 and *p* = 0.01, respectively), suggesting a higher number of natural killer (NK) cells, consistent with our previously published study [42]. Finally, antibodies to EBNA1 and viral capsid antigen (VCA) demonstrate that all children enrolled in this study were EBV-seropositive. As expected, Nandi children from the low malaria exposure area had significantly lower antibody levels against malaria antigens, AMA1 (apical membrane antigen 1) and MSP1 (merozoite surface protein 1) compared to the two groups of children who resided in high malaria exposure areas; Kisumu (*p* < 0.0001 and *p* = 0.01, respectively) and eBL children (*p* < 0.0001 and *p* = 0.007, respectively). AMA1 and MSP1 proteins were gifts from Evelina Angov (Walter Reed Army Institute of Research, Silver Spring, MD, USA).

### 2.3. T Cell Immunophenotyping by Flow Cytometry

PBMCs were thawed and washed twice in complete media (RPMI 1640, GIBCO with 10% heat-inactivated and filter-sterilized fetal bovine serum, Glutamine 1X GIBCO 100 U/mL, Penicillin 100 mg/mL, Streptomycin 1X GIBCO, Hepes BioWhittaker 10 mM) at 37 °C. Live cells were counted using Propidium Iodide (PI) and MACSquant Flow cytometer before overnight incubation (37 °C, 5% CO_2_). Cells were re-counted the next morning, stimulating an average of 250,000 viable cells (minimum of 85% viability) with each one of the following conditions at final concentrations indicated in parenthesis: overlapping EBNA1 peptides (5.76 μM), EBNA2 (10 μg/mL), EBNA3A (10 μg/mL) [43], DMSO (0.13% as a negative control to assess background signal for each individual), or recombinant malaria antigen, *Pf*SEA-1A [38] (5 μg/mL) for 24 h at 37 °C in 5% CO_2_. Cell viability did not differ after stimulation compared to unstimulated cells. We defined individuals as responders when they had higher cytokine expression after antigen stimulation compared to their DMSO/negative control.

Cells were gated on SSC-A vs. FSC-A and single-cell cytoplots (Appendix A). Dump channel applied: CD14/CD19-Pacific Blue (RRID:AB_10373537 and RRID:AB_10373689, respectively) and the Live/dead Zombie Violet (BioLegend cat#423114) for selection of live CD14^−^CD19^−^ cells. T cell subsets and phenotypes were defined by the following monoclonal antibodies: CD3 BV605 (RRID:AB_256437), CD4 PerCP (RRID:AB_2028492), CD8 FITC (RRID:AB_314124), CD45RA Alexa Fluor 700 (RRID:AB_493763), PD-1 PE-Cy7 (RRID:AB_2159324), CCR7 APC-eFluor780 (RRID:AB_1518794), IFN-γ PE-Dazzle 594 (RRID:AB_2563627), IL-10 APC (RRID:AB_398582) and IL-17A PE (RRID:AB_961395). Data were acquired on a 5 laser BD LSRII flow cytometer (UMMS Flow Core Facility) with DIVA software, compensation and FMO controls. Using Live/Dead staining, we obtained an average and median of 77.45% and 86.45% live lymphocytes, respectively. Since IL-10 was measured by intracellular staining of T cells, this was assumed to be human IL-10 and not the viral IL-10 homologue. We analyzed PD-1-positive T cells compared to the unstimulated DMSO control for each condition. T cells subsets were defined by expression of CD45RA and CCR7 as follows: terminal effector (T_EMRA_, CD45RA^+^CCR7^−^), naïve-like (T_NAIVE-LIKE_, CD45RA^+^CCR7^+^), central memory (T_CM_, CD45RA^−^CCR7^+^) and effector memory (T_EM_, CD45RA^−^CCR7^−^) [44]. The expected age-dependent biological differences across T cell subsets showed that children had more T_NAIVE-LIKE_ cells compared to adults with more T_EMRA_ and T_EM_ cells (Appendix A).

### 2.4. Statistical Analysis

Flow cytometry data were analyzed with FlowJo software, version 10.7.1 and GraphPad Prism software, version 9.1.2. Assessment of T cell polyfunctionality was determined using Boolean gating in FlowJo including 3 parameters: IFN-γ, IL-10 and IL-17A and analyzed with SPICE v6.0 [45]. Histograms from SPICE 6 showed the mean of mono/poly-functional expression across study groups and each dot represents one sample, a Wilcoxon Rank Sum Test was applied. Radar (also known as flower) plots were generated using the ggplot2 package in R program (v4.1.0, https://cran.r-project.org/web/packages/ggplot2, accessed on 30 June 2021). Categorical data were analyzed using a chi-square test. Because the number of responders within each group was too low to verify the normality of the underlying distributions, we chose to use non-parametric tests, including Mann–Whitney test (for unpaired analysis), Wilcoxon signed-rank test (for paired analysis), or ANOVA (Friedman for paired data and Kruskal–Wallis for unpaired data). All tests were two-tailed and *p* = 0.05 was set as the level of significance. Because of the exploratory nature of the analysis, we did not use any adjustment of the *p* value for multiple comparisons. The test used is indicated in the legend of each figure. Statistical analysis was performed in GraphPad Prism. Results were expressed using the mean with standard deviation (SD) for dot plots, the mean for the diagrams, and exact *p*-values.

## 3. Results

### 3.1. Heterogeneity of EBNA1 and PfSEA-1A Specific T Cell Responses 

To determine if a history of chronic malaria exposure or the development of eBL was associated with T cell polarization, the frequency of IFN-γ, IL-10 and IL-17A CD4^+^ and CD8^+^ T cell responders to EBNA1 were compared across our three study groups. The frequencies of each cytokine response and the significance of the comparisons between the different groups of children are summarized in Table 1. Malaria-exposed Kisumu children were more likely to have IFN-γ^+^ CD4^+^ T cell responses (84%) to EBNA1 compared to Nandi (66%, *p* = 0.005) and eBL (40%, *p* < 0.0001) children; however, the frequency of IFN-γ^+^ CD8^+^ T cell responders did not differ (63%, 58% and 50%, respectively). In contrast, EBNA1 specific CD4^+^ T cell IL-10 responders were more frequent for eBL (73%) patients compared to Kisumu (38%) and Nandi (33%) children (*p* < 0.0001 for both) but did not differ for CD8^+^ T cells (56%, 54% and 52%, respectively). For EBNA1-specific CD4^+^ T cells expressing IL-17A, eBL and Nandi children had similar response rates (66% and 73%, respectively) which were significantly more prevalent than for Kisumu children (46%, *p* = 0.006 and *p* = 0.0002, respectively). However, IL-17A expressing CD8^+^ T cell response rates to EBNA1 were significantly different across the study groups, with eBL (68%) and Nandi (52%) having significantly more responders than the Kisumu children (18%, *p* < 0.0001 for both). Overall, differences in IFN-γ and IL-10 responders between our study groups involved CD4^+^ T cells to EBNA1, except for IL-17A, which differed for both CD4^+^ and CD8^+^ T cells.

To provide a context for the observed differences in the T cell responses to EBNA1 we compared the frequency of responders within each group to a malaria antigen, *Pf*SEA-1A. Nandi children, who resided in a low malaria exposure area, and thus putatively had acute malaria infections and demonstrably low or undetectable EBV loads, had the same frequency of CD4^+^ T cells (Figure 1A) expressing IFN-γ but significantly fewer IL-10 (*p* < 0.0001) and more IL-17A (*p* = 0.001) responders to EBNA1 compared to *Pf*SEA-1A. In contrast, Kisumu children had similar frequencies of cytokine responders to the EBV and malaria antigens used in our assay, while children diagnosed with eBL had significantly more CD4^+^ T cell IL-17A responders to EBNA1 compared to *Pf*SEA-1A (*p* = 0.0004). Conversely, the frequency of CD8^+^ T cell responders were more variable when comparing EBNA1 to *Pf*SEA-1A within each group of children (Figure 1B). All Nandi children had IL-10 CD8^+^ T cell responses to *Pf*SEA-1A, in contrast to only half of them responding to EBNA1 (*p* < 0.0001) even though they were all EBV seropositive, whereas more Nandi children had IFN-γ CD8^+^ T cell responses to EBNA1 compared to *Pf*SEA-1A (*p* = 0.0006). Kisumu children had more IFN-γ CD8^+^ T cell responders to *Pf*SEA-1A compared to EBNA1 (*p* = 0.01), but the eBL children had similarly low levels to both antigens. Kisumu children were also more likely to have IL-17A CD8^+^ T cell responses to *Pf*SEA-1A compared to EBNA1 (*p* < 0.0001) in sharp contrast to eBL children who had more responders to EBNA1 and few to *Pf*SEA-1A (*p* < 0.0001). The IL-10 CD8^+^ T cell responses to EBNA1 and *Pf*SEA-1A also had an inverse detection with fewer Kisumu responders to *Pf*SEA-1A compared to EBNA1 (*p* = 0.0009), yet more eBL responders to *Pf*SEA-1A compared to EBNA1 (*p* = 0.0002). Taken together, if eBL children developed immune modifications to compensate for the loss of IFN-γ responses to EBNA1, IL-17A could play a role that warrants further exploration. In addition, eBL children seem to have developed more IL-17A expressing CD4^+^ and CD8^+^ T cells compared to Kisumu children who also resided in malaria holoendemic areas.

To determine which combinations of cytokine responses to EBNA1 were found within the same individuals and if they differed by malaria exposure or eBL diagnosis, results were summarized by flower plots (Figure 1C). The colored petals show the different combinations of cytokines being expressed and the size of the petal illustrates the frequency of finding those cytokines (alone or together) within individual children. Most Nandi children (33.3%) only had IL-17A expressing EBNA1-specific CD4^+^ T cells, yet 20% had both IFN-γ and IL-17A, and another 20% had both IL-10 and IL-17A expressing CD4^+^ T cells after EBNA1 stimulation. CD8^+^ T cell responses were more heterogeneous with most Nandi children (20%) expressing both IFN-γ and IL-17A to EBNA1. In contrast, 30% of Kisumu children had only IFN-γ expressing EBNA-1-specific CD4^+^ T cells, with 23.5% of children producing both IFN-γ and IL-17A, or IFN-γ and IL-10. However, their CD8^+^ T cell responses were equally prevalent (27.7%) as one of three phenotypes: IFN-γ, IL-10, or expressing both IFN-γ and IL-10. This might reflect a transition from pro-inflammatory to anti-inflammatory CD8^+^ T cell populations depending on the cumulative number of malaria infections experienced by each child. Of the relatively fewer eBL children who had EBNA1-specific CD4^+^ T cell responses, 27% expressed IL-10 and IL-17A and 20% expressed only IL-10, with lower frequencies of responders for other cytokine combinations. As for their CD8^+^ T cells, 25% of eBL children expressed either IL-10 and IL-17A, or IFN-γ and IL-17A to EBNA1, with some minor exceptions. In summary, it would appear that Kisumu children developed more IFN-γ and IL-10 CD4^+^ and CD8^+^ T cell responses to EBNA1 compared to Nandi and eBL children, suggesting their combined importance in limiting EBV-associated oncogenesis. In contrast, adult memory responses to EBNA1 included IFN-γ, IL-10 expressing cells, but also cells expressing IL-17A and those with a dual expression of IL-10 and IFN-γ (Appendix A).

Figure 1D shows the combinations of cytokine responses to *Pf*SEA-1A found within the same individuals and summarized by the flower plots. CD4^+^ and CD8^+^ T cell responses to this malaria antigen were also found to be heterogeneous within individuals across our study populations. Most of the Nandi children generated IL-10 in combination with CD4^+^ and CD8^+^ T cells that expressed IL-17A and/or IFN-γ to *Pf*SEA-1A, whereas Kisumu children relied on CD4^+^ T cells to express IFN-γ alone or in combination with IL-10 and/or IL-17A, and CD8^+^ T cell populations that expressed IFN-γ or IL-17A. These heterogeneous cytokine profiles were in sharp contrast to the eBL children who had predominantly CD4^+^ and CD8^+^ T cells expressing only IL-10 to *Pf*SEA-1A. Together, these findings reinforce the premise that an immune suppressive environment against malaria developed for children diagnosed with eBL, and that their IL-10 production by T cells was elevated against both EBNA1 and *Pf*SEA-1A. 

### 3.2. Magnitude of Cytokine Responses to EBNA1 and PfSEA-1A 

Even though there were significant differences in the frequency of IFN-γ expressing CD4^+^ T cell responders to EBNA1 across the study groups, we did not observe differences in the magnitude of EBNA1-specific IFN-γ expressing CD4^+^ or CD8^+^ T cells (Figure 2A) for those with responses. To determine if these differences were specific to EBNA1, we tested memory responses to other EBV latent antigens. No differences in IFN-γ responses were observed for EBNA2 and EBNA3A demonstrating that there was no immunodominant latent antigen T cell response within our study populations (Appendix A, respectively).

For *Pf*SEA-1A, we observed significantly fewer IFN-γ expressing CD8^+^ T cells within Nandi children compared to the Kisumu and eBL children (*p* = 0.01 and *p* = 0.03, respectively, Figure 2A). This was expected as Kisumu and eBL children resided in high *Pf*-malaria transmission areas whereas Nandi children had fewer opportunities to develop memory T cell responses to a malaria antigen. However, all children in our study (Nandi, Kisumu and eBL) were EBV seropositive (Appendix A) and should have developed T cell immunity to EBNA1. Therefore, it was not surprising to observe significantly higher frequencies of EBNA1-specific IFN-γ expressing CD8^+^ T cells compared to *Pf*SEA-1A for Nandi children (*p* = 0.01, Figure 2B), whereas Kisumu children had similar frequencies of responses for both antigens. In contrast, eBL children had significantly higher EBNA1-specific IFN-γ-producing CD8^+^ T cells compared to *Pf*SEA-1A (*p* = 0.01, Figure 2B) which suggests a loss of CD8^+^ T cells which could help clear malaria infections. 

Consistent with the frequency of EBNA1-specific IL-10 responders, we did not observe any differences in the frequency of IL-10 expressing CD4^+^ and CD8^+^ T cells across study groups (Figure 2C). However, we found significantly more IL-10-producing CD4^+^ T cells to *Pf*SEA-1A for Nandi compared to eBL children (*p* = 0.02, Figure 2C), with a similar trend for CD8^+^ T cells (*p* = 0.05). We also observed significantly higher frequencies of IL-10 expressing CD4^+^ and CD8^+^ T cells to *Pf*SEA-1A compared to EBNA1 for Nandi children (*p* = 0.04 and *p* = 0.0006, respectively, Figure 2D). IL-10 responses to these antigens were not different for Kisumu or eBL children (Appendix A). Finally, even though Kisumu children did not have significantly higher frequencies of IL-10 expressing CD4^+^ T cells, they had significantly more cytokine production per T cell compared to Nandi and eBL children (*p* = 0.04 and *p* = 0.02, respectively, Figure 2E). Similar comparisons were made for IL-17A responses but no differences were observed (Appendix A). Interestingly, CD4^+^ and CD8^+^ T cell responses from eBL children had a lower IL-17A expression to *Pf*SEA-1A (Appendix A); however, the overall response rates were too infrequent to achieve significance. 

These results demonstrate that the frequency of EBNA1-specific IFN-γ expressing CD4^+^ and CD8^+^ T cells did not differ between eBL and healthy children who developed these T cell responses, suggesting that the IFN-γ deficiency previously reported for eBL children [18] might not be due to the lack of cytotoxic T cells. These findings also suggest that Nandi children who experience acute malaria infections develop robust anti-inflammatory T cell responses to *Pf*SEA-1A mediated by IL-10, in contrast to children who experience malaria as a chronic infection. 

### 3.3. Mono- and Poly-Functional IFN-γ, IL-10 and IL-17A T Cell Responses to EBNA1 and PfSEA-1A

Polyfunctional T cells have been described for other chronic infections and shown to correlate with T cell efficiency and immune regulation [30,46]. However, we found EBNA1-specific CD4^+^ and CD8^+^ T cells to be mostly monofunctional, expressing IL-10 for our three groups of children (Figure 3A,B), thus supporting their role as antiviral, immunosuppressive mediators [47]. Interestingly, IL-17A expressing monofunctional CD4^+^ and CD8^+^ T cells were present in all three groups of children (Figure 3C,D) with a slightly higher trend for eBL compared to Kisumu children (*p* = 0.04, Figure 3C). We also observed a significantly higher frequency of double positive IFN-γ^+^IL-10^+^ EBNA1-specific CD4^+^ T cells for Kisumu compared to eBL children (*p* = 0.02, Figure 3C) identifying a T cell population engaged in immunity to EBV not found in children with eBL. In contrast, no significant differences were observed within EBNA1-specific CD8^+^ T cell profiles across our groups of children despite a trend of higher frequencies of EBNA1-specific CD8^+^ T cells expressing IL-17A from eBL compared to Kisumu children.

Monofunctional *Pf*SEA-1A-specific IL-10-producing CD4^+^ and CD8^+^ T cells were the predominant phenotype for Nandi children compared to eBL children (*p* = 0.01 for both, Figure 4A,B). Polyfunctional T cells were observed at higher frequencies for Kisumu children, with more double positive IFN-γ^+^IL-10^+^
*Pf*SEA-1A-specific CD4^+^ T cells compared to eBL and Nandi children (Figure 4C, *p* = 0.003 and *p* = 0.02, respectively); and more IFN-γ^+^IL-17A^+^
*Pf*SEA-1A-specific CD8^+^ T cells compared to eBL children (Figure 4D, *p* = 0.03). In contrast, eBL children had a higher frequency of IFN-γ^+^-*Pf*SEA-1A-specific CD8^+^ T cells compared to Nandi children with low malaria exposure. Together, these data suggest that separate T cell populations serve distinct roles in fighting infection versus limiting immunopathology. 

### 3.4. EBNA-1 and PfSEA-1A Specific T Cell Subsets

We next determined the effector memory phenotypes of cytokine-producing T cells characterized based on their CD45RA and CCR7 expression and compared subset proportions across our three groups of children and to African adults. EBNA1-specific IFN-γ-producing CD4^+^ T cells from both children and adults were predominantly T_NAIVE-LIKE_ cells with smaller contributions from the other subsets (Figure 5A). Kisumu children displayed a significantly different distribution of EBNA1-specific IFN-γ CD4^+^ T cell subsets compared to Nandi, eBL and adults (*p* = 0.005, *p* = 0.0003 and *p* < 0.0001, respectively) with fewer T_EMRA_ and T_EM_ and no T_CM_. In contrast, EBNA1-specific, IL-10-producing CD4^+^ T cells (Figure 5A) had a larger proportion of T_EMRA_ across our study groups, except for the eBL children who had significantly more T_NAIVE-LIKE_ cells compared to Kisumu children and adults (*p* = 0.03). Additionally, EBNA1-specific, IFN-γ-producing CD8^+^ T cells (Figure 5B) were derived from both T_NAIVE-LIKE_ and T_EMRA_ cell subsets, and displayed significant differences in relative proportions between the Nandi children who had more T_CM_ compared to the Kisumu children, eBL and adult groups, (*p* = 0.001, *p* = 0.001 and *p* < 0.0001, respectively). Interestingly, we observed that IL-10-producing CD8^+^ T cells (Figure 5B) were mostly T_EMRA_ and T_NAIVE-LIKE_ cell subsets, except for the eBL children who had significantly more T_NAIVE-LIKE_ cells compared to the Nandi children and adults (*p* = 0.0005 and *p* = 0.001, respectively) suggesting an immature EBNA1-specific T cell signature for children with this EBV-associated cancer. Further study is needed to determine if this condition precedes oncogenesis or if it is an indication of recent T cell turnover.

As a basis for comparison to persistent EBV latent antigen stimulation, we examined the T cell memory phenotypes to a malaria antigen, which tends to cause an acute infection for Nandi children but a repeated/chronic infection for Kisumu and eBL children. In general, the composition of CD4^+^ T cell subsets expressing IFN-γ and IL-10 to *Pf*SEA-1A were significantly more mature for adults (T_EMRA_, T_CM_ and T_EM_ subsets) compared to Nandi, Kisumu and eBL children (Figure 5C). However, eBL children had more *Pf*SEA-1A-specific IFN-γ CD4^+^ T_EMRA_ and fewer T_CM_ compared to Nandi children (*p* < 0.0001, Figure 5C) and more T_EMRA_ and T_EM_ compared to Kisumu children whose responses were dominated by the T_NAIVE-LIKE_ cell subset (*p* = 0.0005). In contrast, *Pf*SEA-1A-specific IFN-γ CD8^+^ T cells subset proportions were significantly different between malaria-exposed and non-exposed individuals. Nandi children had mostly T_NAIVE-LIKE_ cells with a complete absence of T_CM_ and T_EM_ subsets compared to Kisumu, eBL and adults who resided in malaria holoendemic areas (Figure 5D). Interestingly, we observed no statistical differences for *Pf*SEA-1A-specific IFN-γ CD8^+^ T subsets between eBL children and adults, whereas Kisumu children had more T_NAIVE-LIKE_ and fewer T_EM_ cell subsets compared to adults (*p* = 0.01, Figure 5D). This suggests that eBL children had a more “adult-like” T cell memory repertoire to malaria which could have resulted from a greater cumulative burden of malaria compared to their healthy counterparts. In contrast, differences for *Pf*SEA-1A-specific IL-10-producing CD8^+^ T cell subsets were only apparent when comparing adults to children. IL-10 responses in children were dominated by T_NAIVE-LIKE_ cells, whereas an adult response was governed by T_EMRA_ cells (*p* < 0.0001, Figure 5D). 

Overall, our findings suggest that the compositions of EBNA1 and *Pf*SEA-1A-specific IFN-γ expressing T cell memory subsets were influenced by intensity of malaria exposure, whereas T cell subsets expressing IL-10 were dependent on an eBL diagnosis or age.

### 3.5. PD-1 Expression, as a Marker of Activation or Exhaustion

To determine if T cell exhaustion could account for the differences in the phenotypes and functions described above, we measured the PD-1 expression on CD4^+^ and CD8^+^ T cell subsets within our study groups. PD-1 is a classical T cell exhaustion marker but is also an indicator of the activation and functional adaptation after chronic antigen-engagement [48]. We found that EBNA1 stimulation significantly increased the percentage of PD-1 expression on CD4^+^ and CD8^+^ cells for Nandi children compared to their resting cells (*p* = 0.01 and *p* = 0.002, respectively, Figure 6A,B). Moreover, PD-1 expression, in response to EBNA1, was significantly higher compared to *Pf*SEA-1A-stimulated cells (*p* = 0.005, Figure 6B), as expected for children with acute exposure to malaria. However, both EBNA1 and *Pf*SEA-1A stimulation displayed significant increases in PD-1 expression for CD4^+^ T cells for Kisumu children compared to their resting cells, (*p* = 0.003 and *p* = 0.0005, respectively), and an equal level of induction was found between these two simulation conditions (Figure 6C). Interestingly, *Pf*SEA-1A significantly induced PD-1^+^ expression in CD8^+^ T cells for Kisumu children compared to resting and EBNA1 stimulation levels (Figure 6D). For the eBL children, only *Pf*SEA-1A stimulation increased the percentage of PD-1^+^CD4^+^ cells compared to the resting cells (*p* = 0.002, Figure 6E). However, PD-1 expression was the same as upon EBNA1 stimulation, suggesting a sustained expression for CD4^+^ T cells, a sign of exhaustion as opposed to the transient activation for children with eBL. The PD-1 expression pattern for CD8^+^ T cells was similar between eBL and Kisumu children with higher expressions after *Pf*SEA-1A stimulation, compared to both the resting state and EBNA1 stimulation (*p* = 0.003 and *p* = 0.0005, respectively, Figure 6F). In addition, we found significantly higher overall PD-1 expression on CD4^+^ but not CD8^+^ T cells (Figure 7A,B) for eBL, compared to Nandi children after EBNA1 stimulation (*p* = 0.004). In a striking contrast to the cytokine-producing T cells, the exhausted CD4^+^ T cells were predominantly found in the T_EM_ subset across study groups (including adults) with slightly more PD-1 expressed by T_NAIVE-LIKE_ cells in Kisumu compared to eBL children and adults (*p* = 0.04 and *p* = 0.01, respectively, Figure 7C). However, PD-1 was highly expressed on CD8^+^ T_EMRA_, T_EM_ and T_NAIVE-LIKE_ cells among Nandi, Kisumu and eBL children which differed from adults, who had fewer PD-1 expressing T_NAIVE-LIKE_ cells (*p* = 0.009, *p* < 0.0001 and *p* = 0.003, respectively, Figure 7C). 

We observed similar patterns for *Pf*SEA-1A stimulation with an overall increase in PD-1 expression in CD4^+^ T cells for eBL patients compared to Nandi children (*p* < 0.0001, Figure 7D), and a higher PD-1 expression in CD8^+^ T cells from Kisumu compared to Nandi children (*p* = 0.03, Figure 7E). This result reinforces the notion that PD-1 expression is context-dependent and might be a marker of activation rather than exhaustion. The CD4^+^ T cells expressing PD-1 after *Pf*SEA-1A stimulation were also predominantly T_EM_ and T_CM_ cells (Figure 7F), but showed significant differences when comparing Kisumu children to adults (*p* = 0.02). More differences were observed for CD8^+^ T cells with adults expressing significantly more PD-1 for the T_EM_ subset compared to children who had more PD-1 expression for T_EMRA_ or T_NAIVE-LIKE_ cells. Of note, PD-1 expressing CD8^+^ T cell subset proportions were also different between the Kisumu and Nandi children after *Pf*SEA-1A stimulation (*p* = 0.03). 

Finally, we found that the frequency of PD-1^+^CD4^+^ T cells in response to EBNA1 was higher for Kenyan adults compared to healthy Nandi and Kisumu children (*p* = 0.002 and *p* = 0.01, respectively, Appendix A). Interestingly, the percentage of PD-1^+^CD4^+^ T cells was not different between eBL children and adults, but differed from Nandi children (*p* = 0.004), which suggests that eBL children had exhausted CD4^+^ T cells due to more encounters with EBNA1. Our hypothesis that prolonged malaria exposure (or untreated chronic infections) as a risk factor for eBL is consistent with our finding that the frequency of exhausted CD4^+^ T cells to *Pf*SEA1-1A were similar between eBL children and Kenyan adults, and were significantly higher compared to healthy Nandi children (*p* < 0.0001 and *p* = 0.0004, respectively, Appendix A). Nandi and Kisumu children also had lower frequencies of PD-1^+^CD8^+^ T cells in response to EBNA1 compared to adults, (*p* = 0.01 and *p* = 0.02, respectively, Appendix A), with eBL children being similar to adults for both EBNA1 and *Pf*SEA1-A1. We found higher PD-1^+^CD8^+^ T cell frequencies in adults compared to both Nandi and Kisumu children (*p* < 0.0001 and *p* = 0.002, respectively, Appendix A), which also differed based on their histories of malaria exposure (*p* = 0.03).

## 4. Discussion

The children diagnosed with eBL are deficient in T cell-mediated immune responses specific for EBNA1, the sole EBV antigen consistently expressed in this B cell tumor [18], and the only viral protein expressed during both lytic and latent cycles. Researchers postulate that the cumulative exposure to *Pf* malaria infections throughout early childhood leads to EBV-specific immune deficiencies that put these children at risk of eBL oncogenesis [3]. However, alternative cytokine profiles and the phenotypes of cytokine-producing cells have never been investigated in this context. Here, we characterized EBNA1-specific IFN-γ, IL-10 and IL-17A T cell responses in eBL children compared to healthy children residing in malaria holoendemic (Kisumu) and hypoendemic (Nandi) regions of Kenya to gain further insights into eBL immunopathology. We found that EBNA1-specific CD4^+^ T cells expressing IFN-γ were mostly T_NAIVE-LIKE_ cells and were less frequent in children with eBL compared to healthy children, yet EBNA1-specific T cell cytokine profiles were highly heterogeneous within individuals and across the study groups. Kisumu children had both CD4^+^ and CD8^+^ EBNA1-specific T cells expressing either IFN-γ or IL-10 alone or in combination, compared to predominantly IL-17A expressing CD4^+^ T cells for Nandi children. Kisumu children were also more likely to have polyfunctional CD4^+^ T cells expressing IFN-γ and IL-10, whereas eBL children had more monofunctional IL-17A producing CD4^+^ T cells to EBNA1. The phenotype of cytokine expressing CD8^+^ T cell subsets was largely composed of the T_NAIVE-LIKE_ and T_EMRA_ subsets, with most differences in proportions associated with immune maturation. The studies of adolescents with infectious mononucleosis predominantly generated EBV-specific CD4^+^ T_EM_ and T_CM_ responses [49,50], consistent with our findings that the quality of EBV-specific T cell immunity is dependent on age. Interestingly, Nandi children had more CD8^+^ T_CM_ cells compared to Kisumu and eBL children suggesting a difference in the EBNA1-specific memory T cell establishment relative to the acute versus chronic malaria infections. 

In our seminal study, we reported a lower frequency of IFN-γ responders to EBNA1 within a group of eBL children compared to Kisumu and Nandi children, in addition to having fewer cytokine-producing cells, as measured by ELISPOT [18]. In this study, using a customized flow cytometry assay, we confirmed the lower frequency of responders within another group of children diagnosed with eBL; however, the frequency of IFN-γ-producing CD8^+^ T cells did not differ compared to healthy children. Other studies have demonstrated the importance of not only cytotoxic T cells but also natural killer (NK) cells in EBV control and the prevention of tumorigenesis [51,52]. We previously demonstrated a deficiency in IFN-γ production by NK cells for eBL compared to healthy children [42]. These NK cells were characterized as dysfunctional ‘chronic-infection induced’ cells in children exposed to holoendemic malaria as well as in those diagnosed with eBL [42]. Future studies are needed to explore malaria-induced mechanisms leading to NK cell dysfunction as a component of eBL immunopathology. A limitation of this study was not having enough cells to measure NKT cells, gamma-delta T cells as other non-classical T cell sources of IFN-γ, which were associated with malaria and eBL in other studies [53,54,55]. 

Overall, we observed the robust production of IL-10 from monofunctional CD4^+^ and CD8^+^ T cells (predominantly T_NAIVE-LIKE_ and T_EMRA_ subsets) for all groups of children in response to both EBNA1 and *Pf*SEA-1A. IL-10 is an anti-inflammatory cytokine that down-regulates IFN-γ [56] and has been associated with immune suppression during severe malaria in children [57]. It is not surprising that these children developed an anti-inflammatory response after the prolonged exposure to this persistent EBV infection and that these cytokines may be generated by different T cell subsets compared to adults. IL-10 responses to EBV and malaria antigen were comparable for Kisumu and eBL children; however, Nandi children had significantly higher CD4^+^ and CD8^+^ T cell IL-10 responses to malaria (typically an acute infection in this study population) compared to EBNA1. Several types of CD4^+^ T cells are known to produce IL-10, such as regulatory T cells (Tregs) which can help reduce inflammation, and thereby reduce host tissue damage during chronic diseases. However, Tregs have been associated with poor outcomes for eBL patients [58], and here we observed the same trend with more EBNA1-specific CD4^+^ T cells (but not CD8+ T cells) producing IL-10 in eBL non-survivor compared to survivors (*p* = 0.04, Appendix A). Morales et al. also observed a higher percentage of circulating IL10^+^CD4^+^ T cells in EBV-positive Hodgkin Lymphoma (HL) patients compared to HL patients without EBV and healthy controls [59]. However, they classified these IL10^+^CD4^+^ T cells as Tr1 (T regulatory type 1) using double staining for ITGA2 and ITGB2 [59]. Future studies are needed to assess a panel of IL-10-producing cell types [60] that may contribute to the immuno-regulatory environment in children infected with malaria and diagnosed with eBL. 

Another mechanism used to down-regulate T cell function is PD-1. However, PD-1 can be considered as both a marker of exhaustion as well as activation depending on the type of infection [18,30]. Our study showed that phenotype of antigen-specific, cytokine-producing T cells were not the same subsets as those expressing PD-1 and that PD-1 expression was context-dependent. Children with eBL showed the most signs of T cell exhaustion; whereas, for healthy children, EBNA1 appeared to activate CD4^+^ T cells. Our results reinforce the theory that chronic *Pf* malaria infections and higher EBV loads could lead to persistent T cell activation and dysfunction. These qualitative differences and heterogeneity in T cell immunity to EBNA1 within individuals and across study groups provide more clues in the causal pathway of eBL. Studies are underway exploring this concept, including more markers and transcription factors that regulate T cell activation and exhaustion [61].

To determine the specificity of immune responses to EBNA1 within our study populations, we performed parallel investigations into the responses to a malaria vaccine candidate, *Pf*SEA-1A. Antibodies against *Pf*SEA-1A appear protective against reinfection and decrease parasite replication *in vitro* [38]. However, naturally acquired T cell responses to this antigen have not yet been characterized. We observed that T cells from Nandi children expressed IL-10, IFN-γ and IL-17A to EBNA1; whereas, they almost exclusively expressed IL-10 to *Pf*SEA-1A. Interestingly, both double positive IFN-γ^+^IL-10^+^ and IFN-γ^+^IL-17A^+^ cells were involved in T cell responses to EBNA1 and *Pf*SEA-1A for Kisumu children. In contrast, eBL children had more monofunctional EBNA1-specific T cell responses, whereas the variety of responses decreased for *Pf*SEA-1A, impaired by an overall higher frequency of PD-1. Figure 8 illustrates the heterogeneity found for EBNA1 and *Pf*SEA-1A cytokine responses from children with divergent malaria infection histories compared to children diagnosed with eBL. Deconvoluting human T cell adaptations, driven by the intensity of malaria exposure combined with high EBV loads, supports the prevailing theory of eBL pathogenesis that involves a multifaceted degradation of immune control over EBV-infected B cells. 

There were several limitations to our study. Investigating human T cell immunity to most infections is challenging due to low frequencies (less than 2%) of cells captured in peripheral blood samples. However, the frequencies of EBV-specific T cells in our study were comparable to other study groups [20]. Our sample size was insufficient to test the association between all the different T cell responses we found and eBL survival. We also made assumptions about the histories of malaria exposure for our study populations, using study site as an epidemiologic surrogate. However, based on our previously published longitudinal studies, we are confident that cumulative malaria exposures and EBV loads were significantly different between Kisumu and Nandi children [8,40,62]. It will be informative to validate our results with another cohort of children from Kenya or from another sub-Saharan country where eBL occurs.

## 5. Conclusions

This study demonstrates that EBNA1-specific T cell responses are highly heterogeneous and are shaped by malaria and EBV-associated immune regulatory adaptations. IL-10-producing EBNA1-specific T cells that were enriched in children with eBL might be involved in preventing efficient immune control of this EBV-associated tumor. Understanding the devolution of immune responses to EBV during malaria co-infections will improve our ability to prevent or cure eBL.

## Figures and Tables

**Figure 1 cancers-13-05375-f001:**
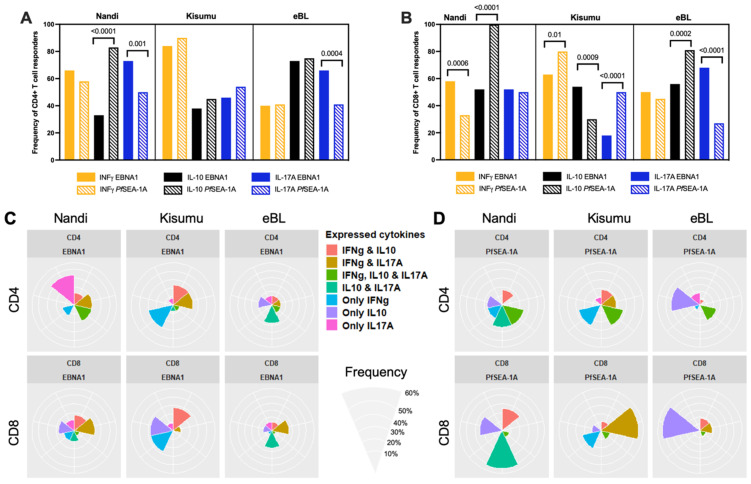
Comparison of cytokine responses for Nandi, Kisumu and eBL children to EBNA1 and *Pf*SEA-1A. Cytokine (IFN-γ, IL-10 or IL-17A) frequencies to EBNA1 and *Pf*SEA-1A were compared for (**A**) CD4^+^ or (**B**) CD8^+^ T cells within each study group, χ^2^ test was applied and *p* values are indicated. Radar/flower plots represent the cytokine profile (frequency from 0 to 50% for Nandi and Kisumu groups and from 0 to 60% for eBL) within individuals from each group of children from CD4^+^ and/or CD8^+^ after (**C**) EBNA1 or (**D**) *Pf*SEA-1A stimulation. Each petal represents a different combination of cytokine (IFN-γ, IL-10, IL-17A) expression found within individual children.

**Figure 2 cancers-13-05375-f002:**
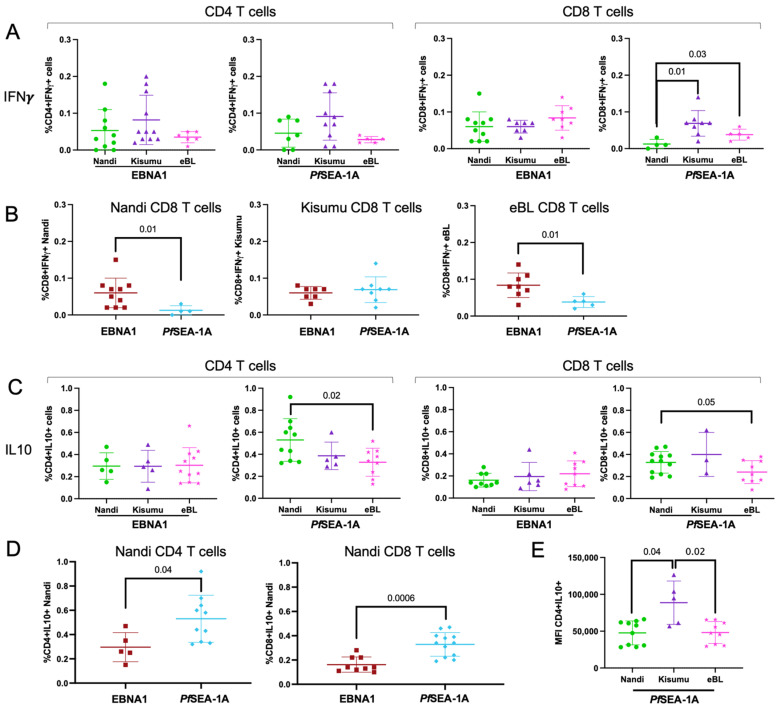
Comparison of cytokine-expressing CD4^+^ and CD8^+^ T cell frequencies across study groups. (**A**) EBNA1 and *Pf*SEA-1A responses were measured as a percentage of total CD4^+^ and CD8^+^ cells expressing IFN-γ and compared across the different groups of children: Nandi (green circle), Kisumu (purple triangle) and eBL (pink star). (**B**) IFN-γ^+^ CD8^+^ T cell responses to EBNA1 (red square) and *Pf*SEA-1A (blue diamond) were compared within Nandi (left), Kisumu (middle), and eBL (right) children. (**C**) EBNA1 and *Pf*SEA-1A responses were measured as the percentage of total CD4^+^ and CD8^+^ T cells expressing IL-10 and compared across the different groups of children: Nandi, Kisumu and eBL. (**D**) IL-10 response to EBNA1 and *Pf*SEA-1A were compared for CD4^+^ (left) and CD8^+^ (right) T cells from Nandi children. (**E**) Comparison of IL-10^+^ expression intensity from CD4^+^ T cells (geometric mean fluorescent intensity, MFI) after *Pf*SEA-1A stimulation across groups of children: Nandi, Kisumu and eBL. Mann–Whitney or Wilcoxon *p*-values are indicated as well as the mean with standard deviation (SD).

**Figure 3 cancers-13-05375-f003:**
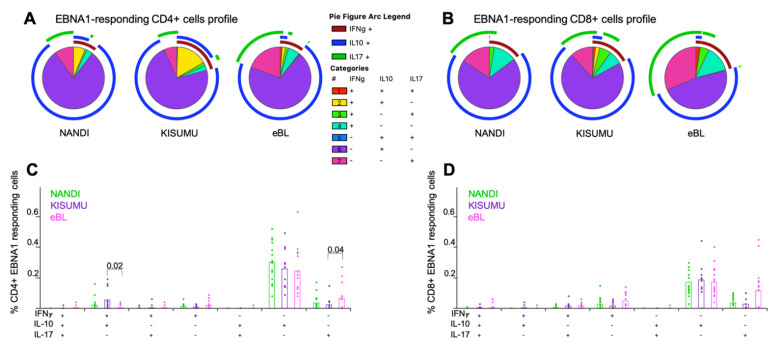
Mono- and poly-functional T cell profiles to EBNA1. Functionality of cells after EBNA1 stimulation were compared across study groups. Each pie slice represents the type of cytokine expression from (**A**) CD4^+^ and (**B**) CD8^+^ T cells; arcs show co-cytokine expression. The frequencies of each type of cytokine-expressing T cells were compared across study groups (Nandi in green, Kisumu in purple and eBL in pink) after EBNA1 stimulation for (**C**) CD4^+^ and (**D**) CD8^+^ T cells. Bar plots show the mean for each group of children and each dot represents an individual. Wilcoxon Rank Sum Test *p* values < 0.05 are indicated.

**Figure 4 cancers-13-05375-f004:**
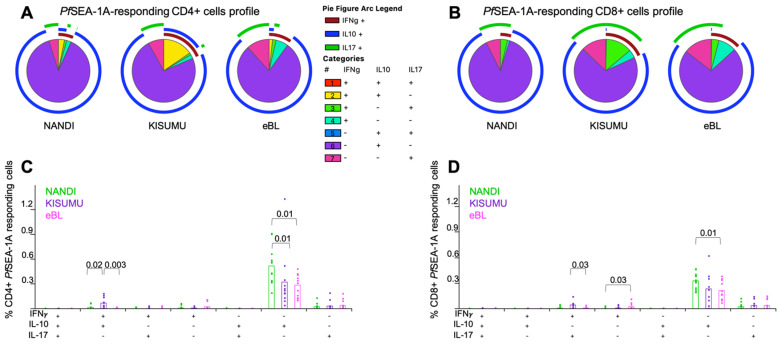
Mono- and poly-functional T cell profiles to *Pf*SEA-1A. Functionality of cells after *Pf*SEA-1A stimulation were compared across study groups. Each pie slice represents the type of cytokine expression from (**A**) CD4^+^ and (**B**) CD8^+^ T cells; arcs show co-cytokine expression. The frequencies of each type of cytokine-expressing T cells were compared across study groups (Nandi in green, Kisumu in purple and eBL in pink) after *Pf*SEA-1A stimulation for (**C**) CD4^+^ and (**D**) CD8^+^ T cells. Bar plots show the mean for each group of children and each dot represents an individual. Wilcoxon Rank Sum Test *p* values < 0.05 are indicated.

**Figure 5 cancers-13-05375-f005:**
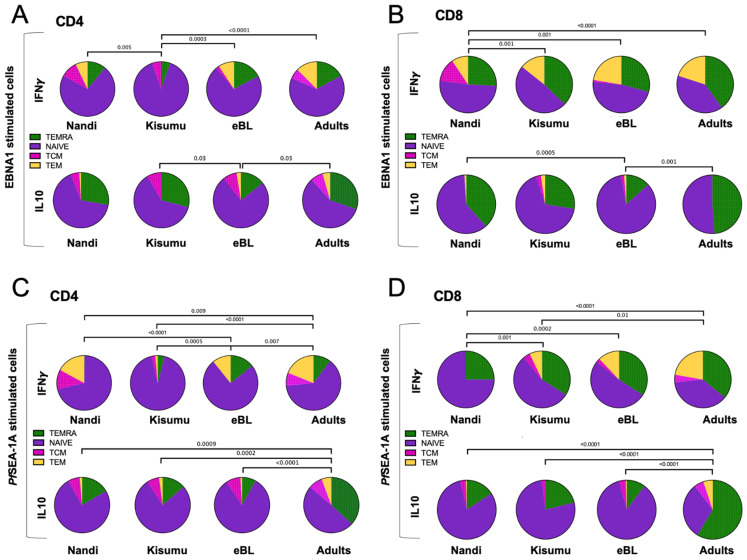
Comparison of EBNA1- or *Pf*SEA-1A- cytokine-expressing T cell subsets across groups of individuals. The mean proportion (pie slice) of IFN-γ or IL-10 expressing (**A**) CD4^+^ and (**B**) CD8^+^ after EBNA1 stimulation; or IFN-γ or IL-10 expressing (**C**) CD4^+^ and (**D**) CD8^+^ after *Pf*SEA-1A stimulation were compared across our study groups (Nandi, Kisumu, eBL, and Adults). T cell subsets were defined by CD45RA and CCR7 expression: T_EMRA_ (green, CD45RA^+^CCR7^−^), T_NAIVE-LIKE_ (purple, CD45RA^+^CCR7^+^), T_CM_ (pink, CD45RA^−^CCR7^+^) and T_EM_ (yellow, CD45RA^−^CCR7^−^). χ^2^ test was applied and *p* values are indicated.

**Figure 6 cancers-13-05375-f006:**
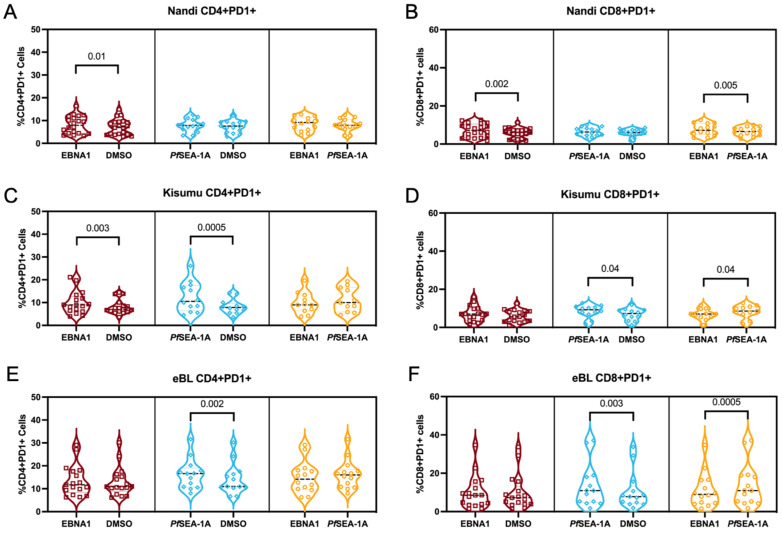
PD-1 expression on stimulated vs. unstimulated CD4^+^ and CD8^+^ T cells. Percentage of CD4^+^PD-1^+^ cells from (**A**) Nandi, (**C**) Kisumu and (**E**) eBL children after EBNA1 and *Pf*SEA-1A stimulation compared to unstimulated, DMSO control. Percentage of CD8^+^PD-1^+^ cells from (**B**) Nandi, (**D**) Kisumu and (**F**) eBL children after EBNA1 and *Pf*SEA-1A stimulation compared to unstimulated, DMSO control. Wilcoxon two-tailed and matched-pairs test *p* values are indicated.

**Figure 7 cancers-13-05375-f007:**
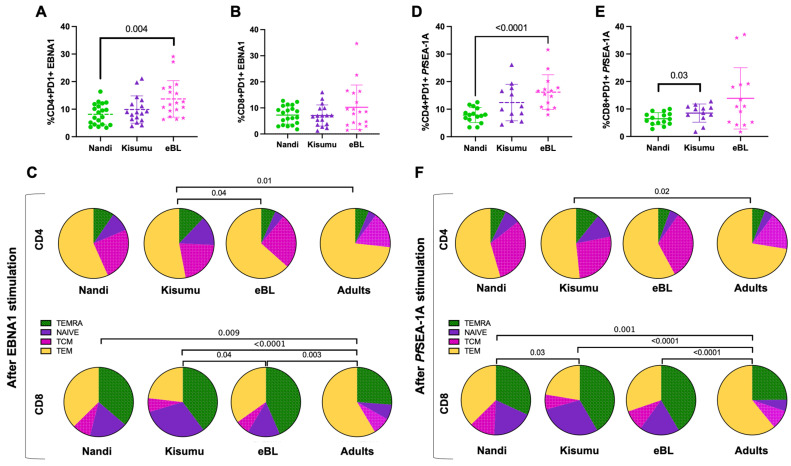
Comparison of PD-1 expressing CD4^+^ and CD8^+^ T cell frequencies across study groups. Comparison of PD-1 expressing (**A**) CD4^+^ and (**B**) CD8^+^ T cells after EBNA1 stimulation between groups of children (Nandi green circles; Kisumu purple triangles and eBL pink stars) using Mann–Whitney test. (**C**) The mean proportion (pie slice) of PD-1-expressing CD4^+^ and CD8^+^ after EBNA1 stimulation was compared across study groups using χ^2^ test. Comparison of PD-1 expressing (**D**) CD4^+^ and (**E**) CD8^+^ T cells after *Pf*SEA-1A stimulation between groups of children (Nandi green circles; Kisumu purple triangles and eBL pink starts) using Mann–Whitney test. (**F**) The mean proportion (pie slice) of PD-1-expressing CD4^+^ and CD8^+^ after *Pf*SEA-1A stimulation was compared across study groups using χ^2^ test. T cell subsets were defined by CD45RA and CCR7 expressions: T_EMRA_ (green, CD45RA^+^CCR7^−^), T_NAIVE-LIKE_ (purple, CD45RA^+^CCR7^+^), T_CM_ (pink, CD45RA^−^CCR7^+^) and T_EM_ (yellow, CD45RA^−^CCR7^−^).

**Figure 8 cancers-13-05375-f008:**
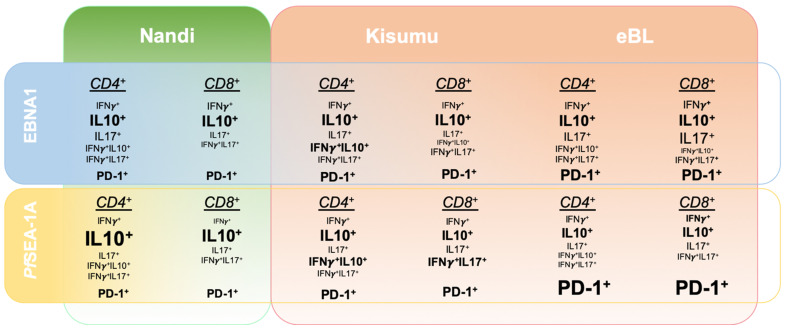
Summary of relative cytokine expression patterns to EBNA1 compared to a malaria antigen, *Pf*SEA-1A across three groups of children: Nandi, low malaria exposure; Kisumu, high malaria exposure; and eBL cancer patients, high malaria exposure. The summary is based on the 2D gating strategy from FlowJo and unbiased Boolean gating in SPICE.

**Table 1 cancers-13-05375-t001:** The frequency of cytokine responders for EBNA1 and *Pf*SEA-1A across study groups. A responder was defined as a child whose T cells produced significantly higher levels of cytokine after ex vivo antigen stimulation compared to their unstimulated, DMSO control cells using chi-square statistics. The frequency of responders was reported for each study group: Nandi, Kisumu and eBL children, and then compared across groups for each cytokine (i.e., IFN-γ, IL-10 or IL-17A), antigen (EBNA1 or *Pf*SEA-1A) and type of T cell response (CD4^+^ and CD8^+^) by Fisher’s exact test; *p*-values < 0.05 were considered significant.

Cytokine	Antigen Stimulation	T Cell Type	Nandi	Kisumu *n* (%)	eBL *n* (%)	Nandi vs. Kisumu (*p* Value)	Nandi vs. eBL (*p* Value)	Kisumu vs. eBL (*p* Value)
IFN-γ responders	EBNA1	CD4	10/15 (66%)	11/13 (84%)	6/15 (40%)	0.005	0.0004	<0.0001
		CD8	10/17 (58%)	7/11 (63%)	8/16 (50%)	0.56	0.32	0.008
	*Pf*SEA-1A	CD4	7/12 (58%)	10/11 (90%)	5/12 (41%)	<0.0001	0.02	<0.0001
		CD8	4/12 (33%)	8/10 (80%)	5/11 (45%)	<0.0001	0.11	<0.0001
IL-10 responders	EBNA1	CD4	5/15 (33%)	5/13 (38%)	11/15 (73%)	0.55	<0.0001	<0.0001
		CD8	9/17 (52%)	6/11 (54%)	9/16 (56%)	0.88	0.67	0.88
	*Pf*SEA-1A	CD4	10/12 (83%)	5/11 (45%)	9/12 (75%)	<0.0001	0.22	<0.0001
		CD8	12/12 (100%)	3/10 (30%)	9/11 (81%)	<0.0001	<0.0001	<0.0001
IL-17 responders	EBNA1	CD4	11/15 (73%)	6/13 (46%)	10/15 (66%)	0.0002	0.35	0.006
		CD8	9/17 (52%)	2/11 (18%)	11/16 (68%)	<0.0001	0.03	<0.0001
	*Pf*SEA-1A	CD4	6/12 (50%)	6/11 (54%)	5/12 (41%)	0.67	0.25	0.08
		CD8	6/12 (50%)	5/10 (50%)	3/11 (27%)	>0.99	0.001	0.001

## Data Availability

The data presented in this study are openly available upon request.

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
