# Peer review of "Interplay between IL-10, IFN-γ, IL-17A and PD-1 Expressing EBNA1-Specific CD4^+^ and CD8^+^ T Cell Responses in the Etiologic Pathway to Endemic Burkitt Lymphoma"

_cancers, 2021, doi:10.3390/cancers13215375_

Round 1

Reviewer 1 Report

Great and timely revisions. Recommend accept.

Reviewer 2 Report

Revisions have adequately responded to reviewers concerns.

This manuscript is a resubmission of an earlier submission. The following is a list of the peer review reports and author responses from that submission.

Round 1

Reviewer 1 Report

This well-written manuscript explores T cell responses to Epstein Barr virus and Plasmodium falciparum antigens in children exposed to hypoendemic and holoendemic malaria, with a group in the latter setting with endemic Burkitts lymphoma, and with inclusion of malaria-exposed adults in some experiments. A flow cyometric approach evaluates cytokine responses and expression of markers of memory and exhaustion in cryopreserved peripheral blood mononuclear cells. The data are comprehensively presented and appropriately contextualized in the existing literature. Statistical analyses are inappropriately applied in some cases, and hypotheses being tested are not consistently or clearly articulated. Overall, the results represent a significant advance in understanding of T cell-mediated responses in malaria and eBL and should be of interest to those in the cancer and malaria fields.

Major comments

Chronicity of infection is referred to repeatedly in this paper, yet no data supporting this is shown. While it is understood that Kisumu and Nandi overall have different malaria epidemiology, conclusions or speculation about the malaria exposure of individuals (such as in line 241-2 ) should be done with caution.

A lot of data are presented in this paper. It is not always clear what the hypothesis is. Perhaps stating the hypothesis for each figure/table would make this clearer. As an example: why are responses in Nandi children compared to eBL children and why are malaria responses compared to EBV responses (eg in Figure 2)? The former comparisons in some cases elicit conclusions that are not well supported as differences in responses to malaria antigen are not different between Nandi and Kisumu children (e.g., figure 2).

Conclusions are being drawn on extremely low levels of responding cells. This should be contextualized with responses observed with other antigens/in other contexts.

Statistics are not always appropriately applied (eg Figures 3 and 4 and Figure 6/7/S5). Multiple hypotheses are being tested with no P value correction. Multivariate analysis should probably be applied in these cases. Statistics should be reviewed by a statistician.

Minor comments

Supp Figure 1: define acronyms in legend; adult control data are not shown here, why? Keep number of significant figures in reported values consistent

M&M:

2.2.: Claims regarding cell subsets not measured  (NK cells) should be done cautiously. If the study was cross sectional, how can claims of chronic or repeated infections be made? Did any of the children have malaria at enrollment? If so, did this impact the findings?

2.3: Did cells die during stimulation? What proportion of cells were live prior to stimulation?

2.4: how were choices made regarding application of parametric vs nonparametric statistics?

Results

Figure fonts throughout are too small

Figure 1/Table 1: how is a responder defined?

Figure 1A and 1B are confusing: within-site malaria vs EBNA1 comparisons are shown in the figure, yet the text refers to trans-site comparisons. Perhaps refer to Table 1 earlier in the text. The hypotheses being tested in Figure 1A and 1B are never really explained as shown.

Lines 246-9: “Overall, it would appear that Kisumu children with exceedingly high EBV loads developed more IFN-? and IL-10 CD4+ and CD8+ T cell responses to EBNA1 compared to Nandi and eBL children, implicating their combined importance in limiting EBV-associated oncogenesis.”. Those who are not eBL or malaria experts will be confused by this concluding remark. Perhaps is should be relegated to the Discussion where supporting evidence can be provided.

Line 295: why are data in figure 2B “not surprising”? what is the null hypothesis?

Line 534: flavors?

Discussion

A very brief but tantalizing reference to survivors vs non-survivors is made in the Discussion/Figure S6. A more expansive examination of this (as opposed to EBV vs malaria antigen responses, for example) would significantly elevate this paper.

Reviewer 2 Report

The article was very interesting to me.
The authors have beautifully described the story of how the EBV virus is correlated with immune markers in various populations of Africa and the correlation between the virus of EBV to the immune system markers (IFN-gamma, PD-L1). Their statistical analysis was well done. The results were clearly presented. I would recommend is the checking of minor grammar and formatting mistakes.

Reviewer 3 Report

This is an excellent manuscript put into excellent perspective and well-interpreted based on the data presented. This is an important topic, understanding the risk factors and underlying immunology for development of Burkitt in sub-Saharan Africa which may provide insights into how to find patients at increased risk or attempt preventive strategies.

I especially love the summary figure comparing the groups.

Minor comments:

In Methods section, 2.1 Study population--you cite "EBV genome and transcriptome studies" which I don't quite understand how that is involved in the current study or what you all were trying to say with that sentence.

The discussion would benefit from some discussion of strenghts and weaknesses, especially the small sample size from one country that requires additional validation in other sites.
